# Genetic structure and knockdown resistance (*kdr*) mutations in *Aedes albopictus* (Skuse) (Diptera: Culicidae): Implications for dengue fever transmission in southeastern China

**Shi-Yuan Lin, Pei-Ling Ye, Ya-Hui Chen, Nan Zheng, Cheng Wu\*, Li-Hua Xie**⊙\*

Department of Pathogenic Biology, School of Basic Medical Sciences, Fujian Medical University, Fuzhou, Fujian, China

\* wucheng@fjmu.edu.cn (CW); lhxie@fjmu.edu.cn (LX)

## Abstract

Dengue fever, transmitted by *Aedes albopictus* in China, is a major public health issue. The emergence of *kdr* gene mutations in *Ae. albopictus* has reduced the efficacy of insecticide-based control. We investigated the genetic structure of eight *Ae. albopictus* populations from China's southeastern coastal region, analyzing genetic diversity, population structure, and the prevalence of *kdr* mutations in relation to dengue fever incidence. Allelic diversity was moderate, with the number of alleles ranging from 2 to 6 and effective number of alleles from 1.52 to 5.00. Genetic clustering revealed four groups with 0.71% to 1.81% variation, indicating moderate differentiation. The coefficient of genetic differentiation value was 0.07 to 0.18, and Nm values 1.13 to 3.25. Except for Foshan and Sanming, four populations showed deltamethrin resistance, 4.31 to 18.87-fold. The *voltage-gated sodium channel (*VGSC) gene analysis identified non-synonymous mutations, with I1532 mutations absent in Sanming and F1534 present in all. Four populations showed resistance to deltamethrin, with resistance levels varying significantly. The VGSC gene analysis revealed multiple non-synonymous mutations associated with resistance, particularly at positions I1532 and F1534. No significant correlation was found between dengue fever incidence, *kdr* mutations, and genetic indicators, indicating a consistent potential for disease transmission. However, populations with higher genetic diversity had lower frequencies of F1534 and higher frequencies of I1532I mutations. The findings underscore the significant influence of *kdr* mutations on the effectiveness of mosquito control strategies. The presence of these mutations necessitates the implementation of alternative insecticides and integrated pest management approaches to sustainably reduce *Ae. albopictus* populations and mitigate the spread of dengue fever.

## Introduction

Dengue fever poses a significant public health threat worldwide, with effective vaccines and treatments remaining limited [1,2]. In China, *Aedes albopictus* is the primary vector for dengue transmission [3]. The abundance of this species and the incidence of dengue vary across

**Data availability statement:** All relevant data are within the manuscript and its Supporting information files.

**Funding:** This research was supported by Natural Science Foundation of Fujian Province (2023J01543), the Startup Project for High-level Talents of Fujian Medical University (XRCZX2020016),and the National Nature Science Foundation of China (82402662), the Fujian Medical University Qihang Fund (2021QH1005), and the Education and Scientific Research Project for Young and Middle-aged Teachers in Fujian Province (JAT210116).

**Competing interests:** The authors have declared that no competing interests exist.

its geographic distribution[4]. These variations could be attributed to population-specific susceptibilities to the dengue virus [5].

Therefore, effective management of *Ae. albopictus* populations and disruption of their transmission pathways are crucial for dengue prevention strategies. Microsatellite markers are valuable tools for population genetic studies due to their ease of use, detectability, reproducibility, polymorphism, and co-dominant inheritance [6,7]. The TP-M13-SSR technique excels in identifying unique alleles at specific loci, ensuring data standardization across batches, and simplifying the analysis of PCR amplification products [8].

Given the critical role of *Ae. albopictus* in dengue transmission, understanding its population genetics is essential. Moreover, as the primary method of dengue fever prevention and control relies on insecticide application, it is equally important to explore the impact of these interventions on the mosquito population, particularly in terms of developing resistance. Chemical applications, particularly pyrethroids, is the principal strategy for *Ae. albopictus* control. The development of insecticide resistance has complicated vector management. [9,10]. Target site insensitivity is a predominant resistance mechanism in *Ae. albopictus*, often linked to genetic alterations in the *voltage-gated sodium channel (VGSC)* gene [11]. This gene mutation, known as knockdown resistance (*kdr*) [12], results in diminished mosquito response to pyrethroid insecticides. Common mutations identified in field populations include F1534S, F1534C, I1532T, and V1016G [13,14].

The high prevalence of *kdr* mutations in China suggests a rapid adaptive process in response to insecticide pressures. Our previous investigations have revealed a link between the *kdr* mutation profile in *Ae. albopictus* populations of Fujian Province and the levels of deltamethrin resistance encountered at multiple collection points [15]. The mosquito's varying geographical distribution leads to differences in population density and the spread of dengue fever, which may relate to population-specific susceptibility to the virus. This study aimed to investigate the genetic profiles of *Ae. albopictus* populations across eight sampling sites in four provinces in southeast China using TP-M13-SSR technology, identifying correlations between genetic diversity and dengue incidence. Additionally, the study examined the resistance of *Ae. albopictus* strains in five cities in Fujian Province and Foshan, analyzing the genetic correlation between *kdr* mutations and population structure in Fujian Province.

## Materials and methods

### Mosquito sampling

We selected eight *Ae.albopictus* collection sites across diverse habitats in Sanming (SM), Fuzhou (FZ), Putian (PT), Quanzhou (QZ), and Zhangzhou (ZZ) in Fujian Province; Guangzhou (GZ) in Guangdong Province; Ganzhou (JX) in Jiangxi Province; and Hangzhou (ZJ) in Zhejiang Province (Fig 1). Thirty adult *Ae.albopictus* mosquitoes were collected at each sampling site. Each sampling point is divided into 3 groups, with each group containing 10 adult *Ae.albopictus* mosquitoes. Sampling for each group is carried out in different districts of the city. During the collection, larvae of *Ae. albopictus* were collected from various habitats across different cities, such as shade, flowerpot, or green belt. These sites were chosen for their high incidence of dengue fever and represent first-level province for surveillance purposes [16]. Male and female adult mosquitoes were collected for genetic analysis to explore their association with dengue incidence and resistance mutations. Female mosquitoes were fed on commercial defibrinated sheep blood (Hongguan Biotechnology Company) to facilitate egg laying for the subsequent generation's development. Larvae were fed turtle food to the adult stage in the laboratory. The third and fourth larvae(L3-L4) stage of the first and second filial generation(F1-F2) were specifically used for pyrethroid susceptibility testing. A sensitive

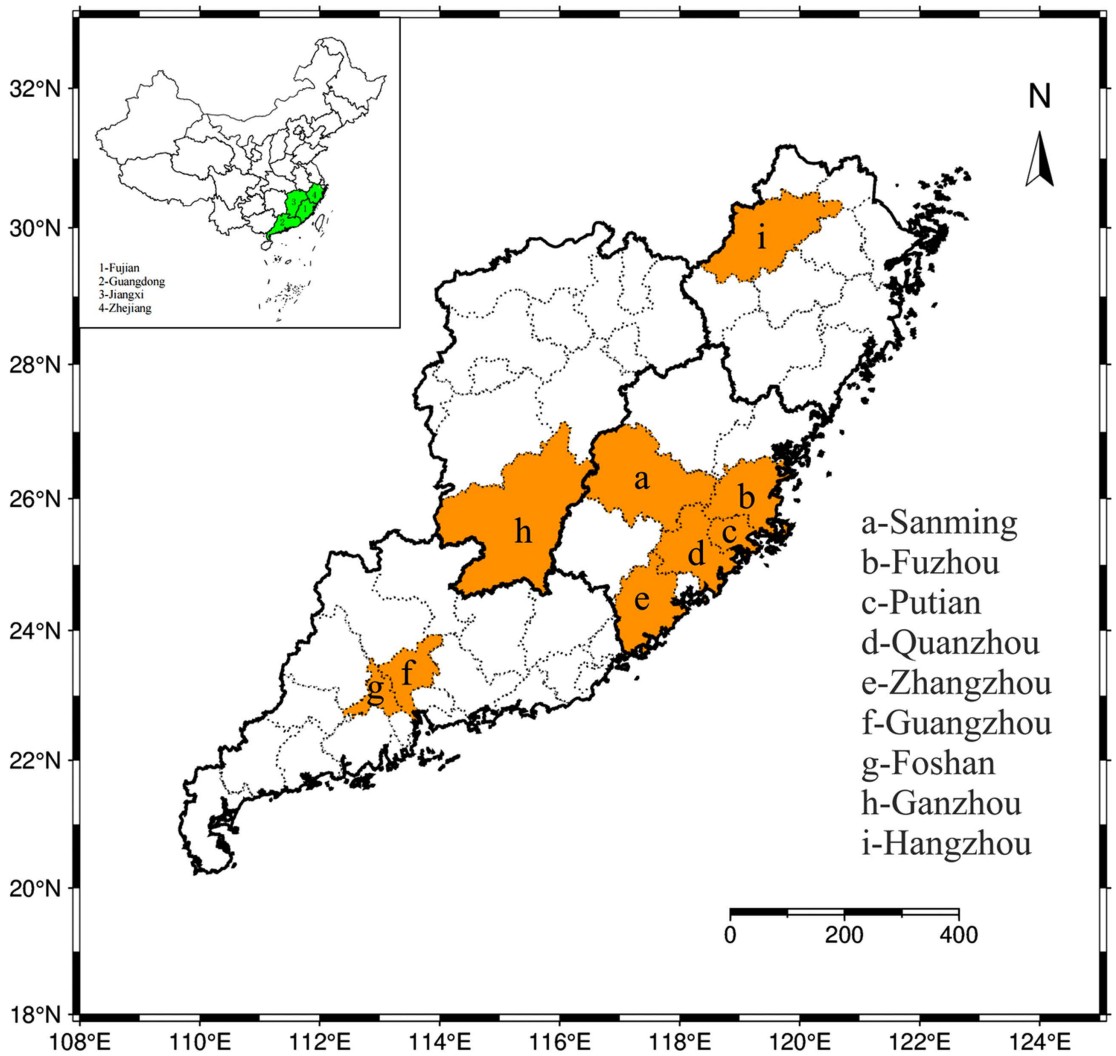

**Fig 1. Map of Sampling Site.**

*Ae.albopictus* Foshan (FS) strain, provided by Professor Chen Xiaoguang's group at Southern Medical University, served as a control. The breeding environment was maintained at $26 \pm 2°C$, $70 \pm 10\%$ relative humidity, and a 16:8 light-dark cycle.

## Amplification and electrophoretic analysis of microsatellite markers

In this study, all DNA was extracted using the Aidlab DNA Rapid Extraction Kit (DN07) and stored at -20°C. Nine microsatellite loci on three chromosomes of *Ae.albopictus* was selected for their ability to reflect genetic diversity and population structure [5]. The microsatellite sequences and their primers are detailed in the accompanying S1 Table. The forward primers of these selected microsatellite sequences were tagged with the universal primer M13, facilitating PCR amplification of *Ae.albopictus* DNA. This process utilized a 25μL reaction system, incorporating the following reagents: ddH$_2$O (Nearshore Protein Technology Co., Ltd.) 3.5μL; Forward primer (Sangon Bio-technology Co. Ltd., 10μmol/L) 0.5μL; Reverse primer (Sangon Biotechnology Co., Ltd., 10μmol/L) 2.5μL; M13 primer (Sangon Biotechnology Co.,

Ltd., 10μmol/L) 2.0μL; sample DNA 4.0μL; and *Taq* polymerase (Baozhi Physics Technology Co. Ltd., 1.25 U/25μL) 12.5μL. The PCR protocol involved an initial denaturation at 94°C for 3 min, followed by 35 cycles of denaturation at 94°C for 30 s, annealing at optimized temperatures for each locus (53°C for SSRmic9,10; 55°C for SSRmic5,11,12,16; 56°C for SSRmic3,6,8) for 30 s, extension at 72°C for 30 s, and a final extension at 72°C for 10 min. Amplification success was confirmed using 1% agarose gel electrophoresis, and samples with distinct bands were sequenced using capillary electrophoresis at Shanghai Sangon Biological Engineering Co., Ltd.

## Determination of resistance to deltamethrin in *Ae.albopictus* larvae

We assessed larval resistance to deltamethrin using the World Health Organization (WHO)-recommended protocol. Probit analysis was employed to calculate the LC50 (the concentration of deltamethrin causing 50% mortality in the larval population). Bioassays were conducted with 25 third or fourth-instar *Ae.albopictus* larvae in 99 mL of dechlorinated tap water, to which 1 mL of insecticide solution at varying concentrations was added. Five concentration gradients were tested for deltamethrin, designed to achieve mortality rates between 10% and 90%, with three replicates per concentration (S1 Appendix, Excel. S1 Bioassay Experiments). Mortality was assessed after 24 hours, and the LC50 was determined.

## Death determination and correction for interference factors

Dead larvae or pupae were identified by their lack of response to gentle probing with a suction pipette. If more than 10% of the control larvae pupate in the course of the experiment, the test should be discarded and repeated. If the control mortality is between 5% and 20%, the mortalities of treated groups should be corrected according to Abbott's formula (below).

$$\text{Corrected mortality } (\%) = \frac{\% \text{ mortality in test} - \% \text{ mortality in control}}{100 - \% \text{ mortality in control}} \times 100$$

Resistance ratios (RR) = LC50 (field population)/LCRR (susceptible population)

When RR is < 5 the field population is considered susceptible, when RR is between 5 and 10 mosquitoes are considered to have moderate resistance, and when RR is > 10 the mosquitoes are highly resistant (https://www.who.int/publications/i/item/WHO-ZIKV-VC-16.1).

## Mutation analysis of two representative alleles of the *VGSC* gene in Fujian strains

The mutation rates at codons I1532 and F1534 in the *VGSC* gene of *Ae.albopictus* populations from SM, QZ, ZZ, PT, and FZ cities in Fujian Province were assessed with sample sizes of 30, 26, 32, 30, and 36. Genomic DNA was extracted from a single adult *Ae. albopictus* specimen using a DNA extraction kit (Adlai Biotechnology, Beijing, China). The extracted DNA served as a template for amplifying fragments corresponding to domains II and III of *VGSC*. Primers were designed based on the literature and synthesized for specificity to their target sequences [11]. The PCR reaction mixture, totaling 25μL, contained 12.5μL of PCR Master Mix(*TaKaRa* Taq™ Version 2.0), 2μL each of 10μmol/L forward and reverse primers, 5.5μL of template DNA, and nuclease-free water to complete the volume. The PCR cycling conditions were optimized as follows: an initial denaturation step at 94°C for 3 minutes; followed by 35 cycles of denaturation at 94°C for 30 seconds, annealing at 55°C (for domain II) or 53°C (for domain III) each for 30 seconds, and extension at 72°C for 30 seconds; concluding with a final extension at 72°C for 10 minutes. Post-PCR, products were resolved by 1% agarose gel

electrophoresis, and those displaying clear bands without smear were purified and sequenced by Shangya Company. Subsequent sequence alignment and mutation analysis were performed using MEGA (v.11.0.13) and Chromas (v.2.6.5).

## Data analysis

Utilizing Genemapper, we extracted and analyzed sequence peak maps and microsatellite fragment size data. SSR data were analyzed using GeneMapper software to convert peak plot data into allele lengths for subsequent statistical analyses. Genetic diversity indices, including the number of alleles (NA), effective number of alleles (NE), Shannon index (I), observed heterozygosity (Ho), and expected heterozygosity (He), were calculated by using the GenAlEx 6.503 software; Additionally, inbreeding coefficient (Fis), the total population inbreeding coefficient (Fit), the coefficient of genetic differentiation (Fst), and the gene flow (Nm), were determined to assess population genetic structure and gene exchange. Principal coordinates analysis (PCoA) and analysis of molecular variance (AMOVA) were performed using the same software to further elucidate population genetic relationships. We performed a Pearson correlation analysis to investigate the relationship between six genetic indicators and the incidence of dengue fever in 2019. The genetic indicators included NA, NE, I, Ho, He and the Fis. Also, we conducted a Pearson correlation analysis to assess the relationship between the six genetic indicators and the occurrence of *kdr* mutations in *Ae.albopictus* populations. The genetic indicators included NA, NE, I, Ho, He and the fixation index(F). Both Pearson correlation analyses were performed using IBM SPSS Statistics 28 and the results were considered statistically significant when the p-value < 0.05. Mortality data obtained at each concentration was subjected to probit-regression analysis to determine LC50 valued by SPSS (v.25.0). The graphical representation of the results was prepared using GraphPad Prism software (v.9.0).

## Results

### Capillary electrophoresis test results

Of the evaluated loci, *Aealbmic3, 5, 6, 8, 9*, and *11* were not detected in more than two-thirds of the individuals, Consequently, the genetic analysis focused on the *Aealbmic10, 12*, and *16* loci, which were consistently present. This study focused on the Fuzhou (FZ) population, showcasing the peak maps of the nine microsatellite loci (S1 Fig).

### Diversity of i*ndividual Ae.albopictus* populations

The genetic analysis of the *Ae.albopictus* population was based on the three screened SSR loci (*Aealbmic10, 12,* and *16*). Overall, the microsatellite loci characteristics of the 8 sampling sites are relatively close, and the microsatellite loci have a certain degree of polymorphism in each population. The genetic indices for each locus, including the polymorphic information content and the Shannon index, indicated high levels of polymorphism (Table 1). The NA of per population ranged from 2 to 6, the NE varied from 1.52 to 5.00, and the expected heterozygosity (He) spanned from 0.34 to 0.80 (Table 2). Among them, at the *mic10* locus,

Table 1. Genetic Indicators for Each Microsatellite Locus in *Ae.albopictus* Populations.

| Locus | NA | P | I | Ho | He |
|---|---|---|---|---|---|
| Aealbmicl0 | 3.25 | 1.00 | 0.94 | 1.00 | 0.55 |
| Aealbmicl2 | 5.00 | 1.00 | 0.72 | 1.00 | 0.87 |
| Aealbmicl6 | 3.00 | 1.00 | 0.94 | 0.98 | 0.56 |

the NA (6), NE (4.17) and I (1.61) of the Zhangzhou population (ZZ) are all higher than those in other regions; at the *mic12* locus, the NA (6) and NE (5.00) of the Fuzhou population(FZ), Guangzhou population(GZ) and Zhejiang population(ZJ) are higher than those in other regions, and the I (1.70) of the Fuzhou population(FZ) and Zhejiang population(ZZ) is higher than that in other regions; at the mic16 locus, the NA (5), NE (3.13) and I (1.36) of the Guangzhou population(GZ) are all higher than those in other regions, and the diversity of the above-mentioned populations is relatively rich. The strain richness levels in the remaining regions are similar. The Ho of all populations sampled this time is greater than He, indicating a rich heterozygote.

**Table 2. Genetic Indicators for Each *Ae.albopictus* Population.**

| Microsatellite loci | P | N | NA | NE | I | Ho | He |
|---|---|---|---|---|---|---|---|
| Aealbmic10 | SM | 10 | 2 | 1.72 | 0.61 | 1.00 | 0.42 |
| | FZ | 10 | 2 | 1.72 | 0.61 | 1.00 | 0.42 |
| | PT | 10 | 3 | 2.38 | 0.94 | 1.00 | 0.58 |
| | QZ | 10 | 3 | 1.52 | 0.64 | 1.00 | 0.34 |
| | **ZZ** | **10** | **6** | **4.17** | **1.61** | **1.00** | **0.76** |
| | GZ | 10 | 3 | 2.28 | 0.90 | 1.00 | 0.56 |
| | JX | 10 | 4 | 3.33 | 1.28 | 1.00 | 0.70 |
| | ZJ | 10 | 3 | 2.38 | 0.94 | 1.00 | 0.58 |
| Aealbmic12 | SM | 10 | 4 | 2.38 | 1.09 | 1.00 | 0.58 |
| | **FZ** | **10** | **6** | **5.00** | **1.70** | **1.00** | **0.80** |
| | PT | 10 | 4 | 3.57 | 1.31 | 1.00 | 0.72 |
| | QZ | 10 | 4 | 2.38 | 1.09 | 1.00 | 0.58 |
| | ZZ | 10 | 5 | 4.55 | 1.56 | 1.00 | 0.78 |
| | **GZ** | **10** | **6** | **5.00** | **1.67** | **1.00** | **0.80** |
| | JX | 10 | 5 | 3.57 | 1.42 | 1.00 | 0.72 |
| | **ZJ** | **10** | **6** | **5.00** | **1.70** | **1.00** | **0.80** |
| Aealbmic16 | SM | 10 | 3 | 2.63 | 1.03 | 0.80 | 0.62 |
| | FZ | 10 | 3 | 2.17 | 0.90 | 1.00 | 0.54 |
| | PT | 10 | 3 | 1.85 | 0.80 | 1.00 | 0.46 |
| | QZ | 10 | 3 | 2.94 | 1.09 | 1.00 | 0.66 |
| | ZZ | 10 | 2 | 1.72 | 0.61 | 1.00 | 0.42 |
| | **GZ** | **10** | **5** | **3.13** | **1.36** | **1.00** | **0.68** |
| | JX | 10 | 3 | 2.63 | 1.03 | 1.00 | 0.62 |
| | ZJ | 10 | 2 | 2.00 | 0.69 | 1.00 | 0.50 |

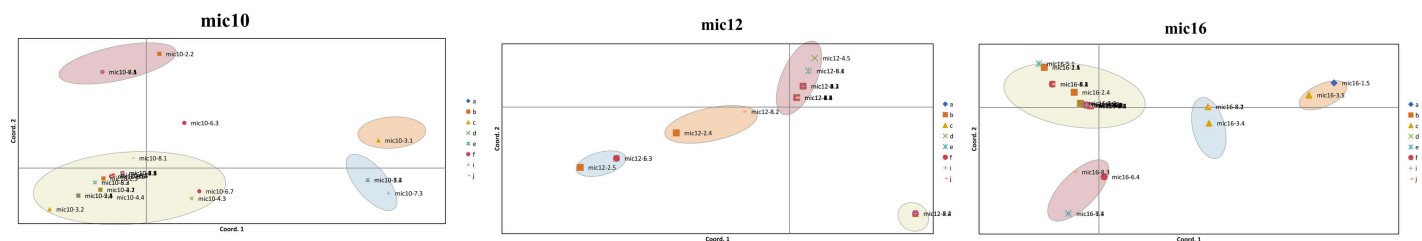

**Fig 2. Plot of PCoA results for the three microsatellite loci.:SM; b: FZ; c:PT; d: QZ; e: ZZ; f: GZ; i: JX; j: ZJ.**

## Analysis of genetic differences between *Ae.albopictus* species

The genetic similarity situation among *Ae.albopictus* populations was assessed using PCoA, based on genetic distances computed from co-dominant genotypes. From the PCoA results, the eight populations were discernibly clustered into four groups (Fig 2). Subsequent analyses included population differentiation, gene flow estimation, and AMOVA. The inbreeding coefficient (Fis) within the populations ranged from -0.57 to -0.38, while the total population inbreeding coefficient (Fit) spanned from -0.56 to -0.13. Considering that for all the populations mentioned above, the observed heterozygosity (Ho) is greater than the expected heterozygosity (He) at the three loci, it can be deduced that the mosquitoes from all sampled locations all result from out - breeding. This leads to a high degree of heterozygosity, which is consistent with the analysis results of the number of alleles (NA), effective number of alleles (NE), and Shannon's information index (I). Generally, an Fst value less than 0.05 is considered to indicate mild population differentiation. When the Fst value ranges from 0.05 to 0.15, it suggests moderate population differentiation. And if the Fst value is between 0.15 and 0.25, it implies high - level population differentiation. The genetic differentiation among populations was measured by Fst. The values ranged from 0.07 to 0.18, falling within the range of 0.05 - 0.25, indicating a moderate to high level of genetic differentiation among the four groups. The estimated Nm values fall between 1.13 and 3.25, suggesting that the populations can withstand the inter - population genetic differentiation resulting from genetic drift and that gene flow exists among them. (shown in Table 3). Results of AMOVA indicated that all the genetic variation was predominantly attributable to differences among individuals within each population (98%～100%), with minimal variance accounted for by divergence between populations (0 to 2%) (S2 Fig).

## Resistance of *Ae. albopictus* to deltamethrin and analysis of the correlation between population genetic indicators and *kdr* mutations in various cities of Fujian Province

Our early results showed that SM strain of *Ae.albopictus* demonstrates sensitivity to deltamethrin. The ZZ and QZ strains display resistance ratios (RR) of about 4.31 and 4.51, respectively, suggesting a level of sensitivity approaching moderate resistance. In contrast, the PT and FZ strains have established a high degree of resistance to deltamethrin, with resistance ratios estimated at approximately 15.80 and 18.87, respectively. $LC_{50}$ represents the concentration of deltamethrin required to cause mortality in 50% of the larvae. The sensitive population of *Ae. albopictus* in FS exhibits an $LC_{50}$ of 2.03 μg/L against deltamethrin. The SM population has an $LC_{50}$ of 3.37 μg/L, QZ population of 9.16 μg/L, ZZ population of 8.75 μg/L, PT population of 33.08 μg/L, and FZ population of 38.31 μg/L(Fig 3). Therefore, it can be seen that the *Ae. albopictus* of the SM population was relatively sensitive to deltamethrin. The resistance levels are shown in Table 4 and Fig 3, with the raw data are included in the S1 Appendix, Excel. S1. Bioassay Experiments.

Non-synonymous mutations were identified within these populations. In the I1532 position, a variety of non-synonymous mutations were observed across all populations except SM.

Table 3. Population differentiation and gene flow analyses for each microsatellite locus.

| Locus | Fis | Fit | Fst | Nm |
|---|---|---|---|---|
| Aealbmicl0 | -0.57 | -0.45 | 0.07 | 3.19 |
| Aealbmicl2 | -0.38 | -0.13 | 0.18 | 1.13 |
| Aealbmicl6 | -0.67 | -0.56 | 0.07 | 3.25 |

In the FZ population, two such mutations were identified: I1532Y and I1532F, with frequencies of 8% and 17%, respectively. At the F1534 position, diverse non-synonymous mutations were noted in all populations. Notably, the PT and FZ populations exhibited the F1534S and F1534L mutations with frequencies of 90% and 33.33%, and 10% and 8.33%, respectively. Table 5 presents the genotype and frequency of the two sites (I1532, and F1534) in the *VGSC* gene of *Ae.albopictus* from various cities in Fujian Province.

Further, we conducted correlation analysis between population genetic indicators and *kdr* mutations in various cities of Fujian province. The result revealed no statistically significant correlations between these genetic indicators and the incidence of *kdr* mutations. Specifically,

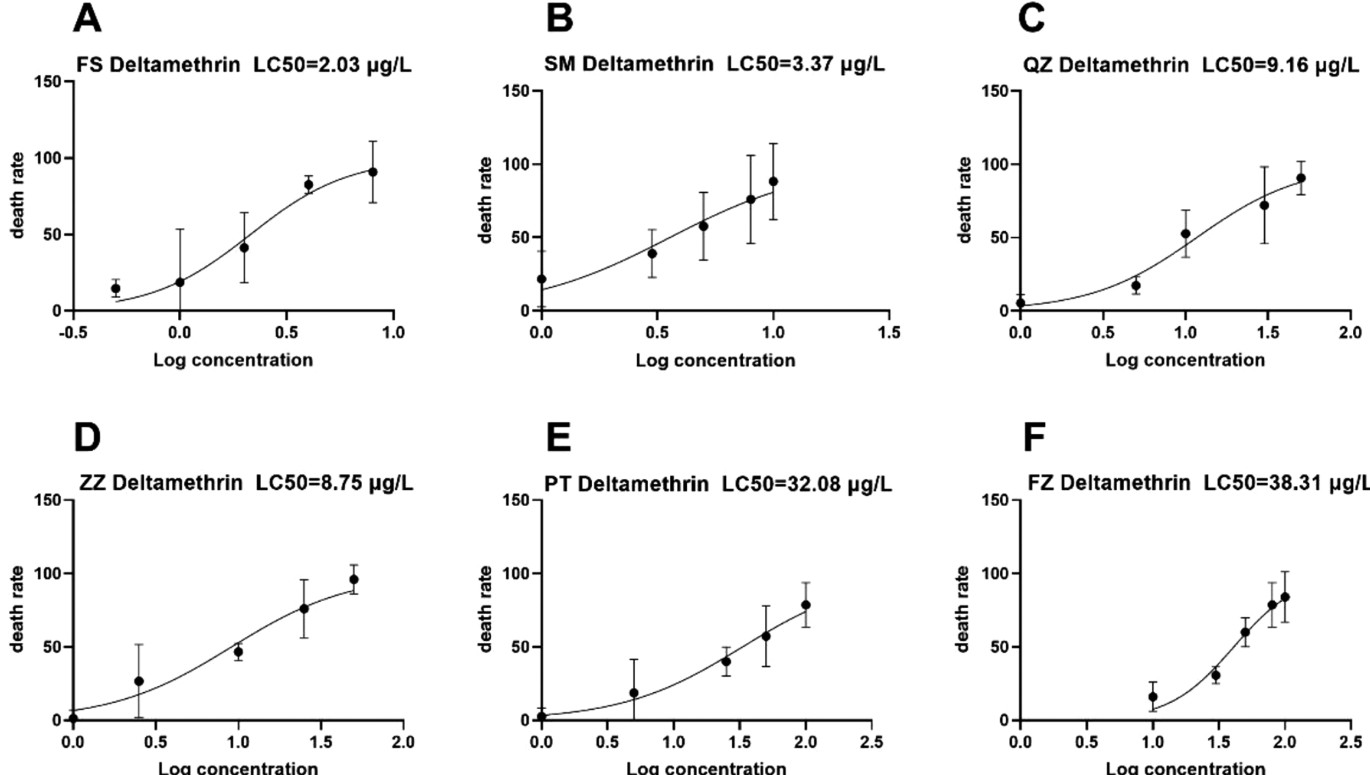

**Fig 3. Resistance of *Aedes albopictus* to deltamethrin.** (A) FS strain. (B) SM strain. (C) QZ strain. (D) ZZ strain. (E) PT strain. (F) FZ strain.

**Table 4. Levels of Resistance of *Ae.albopictus* to Deltamethrin in Various Cities of Fujian Province.**

| Region | deltamethrin | |
|---|---|---|
| | LC$_{50}$ (95%CI) (μg/L) | RR |
| **FS**[*] | **2.03 (1.26, 3.29)** | **1.00** |
| SM | 3.37 (1.77, 5.17) | 1.66 |
| QZ | 9.16 (7.78, 10.71) | 4.51 |
| ZZ | 8.75 (3.23, 20.88) | 4.31 |
| PT | 32.08 (25.56, 40.61) | 15.80 |
| FZ | 38.31 (24.23, 54.33) | 18.87 |

[*]:*FS strain is the reference strain.*

the correlation coefficients (r) and corresponding *p*-values for the number of alleles, effective alleles, Shannon's index, observed heterozygosity, expected heterozygosity, and fixation index were 0.51 *(p = 0.38)*, 0.58 *(p = 0.31)*, 0.58 *(p = 0.30)*, 0.12 *(p = 0.84)*, 0.62 *(p = 0.27)*, and 0.23 *(p = 0.72)*, respectively (Table 6). These findings suggest that the genetic indicators examined do not strongly predict the presence of *kdr* mutations in the populations studied. However, we observed that populations with greater genetic differences (F) tended to have lower frequencies of F1534 and higher frequencies of I1532I (Table 5 and Table 6).

## Analysis of the correlation between *Ae.albopictus* population genetic indicators and the incidence of dengue fever

Moreover, we analyzed the correlation between *Ae.albopictus* population genetic indicators and the incidence of dengue fever. The analysis revealed no statistically significant correlation between these genetic indicators and the incidence of dengue fever, with the top three correlation coefficients being associated with the Shannon index (r = 0.77, *p* = 0.23), the number of alleles (r = 0.74, *p* = 0.26), and expected heterozygosity (r = 0.26, *p* = 0.74) (Table 7).

## Discussion

### Genetic diversity and dengue fever transmission

This study employed TP-M13-SSR technology to investigate the genetic analysis of *Ae.albopictus* populations in several Dengue Fever Surveillance Level I Provinces along China's southeastern coast. Three microsatellite loci, Aealbmic10, 12, and 16, effectively captured the genetic traits and clustering patterns of the collected specimens. High individual genetic diversity was

**Table 5. Genotype and Frequency of Two Sites (I1532, And F1534) in the VGSC Gene of *Ae.albopictus* in Various Cities of Fujian Province.**

| Region | I1532 | | | | F1534 | | | |
|---|---|---|---|---|---|---|---|---|
| | I1532I | I1532Y | I1532T | I1532F | F1534F | F1534S | F1534L | F1534C |
| SM | 30(100%) | / | / | / | 18(60%) | 9(30%) | 3(10%) | / |
| QZ | 10(38.46%) | 10(38.46%) | 2(7.69%) | 4(15.38%) | / | 22(84.62%) | 2(7.69%) | 2 (7.69%) |
| ZZ | 28(87.5%) | 4(12.5%) | / | / | 24(75%) | 8(25%) | / | / |
| PT | 27(90%) | / | / | 3(10%) | / | 27(90%) | 3(10%) | / |
| FZ | 27(75%) | 3(8%) | / | 6(17%) | 21(58.33%) | 12(33.33%) | 3(8.33%) | / |

Note: "/" indicates the absence of mutation at that site; I1532I, I1532Y, I1532T, I1532F, F1534F, F1534S, F1534L, F1534C represent genotypes, where I, Y, T, F, S, L denote isoleucine, tyrosine, threonine, phenylalanine, serine, and leucine, respectively. The data in the table show the frequencies of each genotype corresponding to different genotypes at each mutation site. Numbers in parentheses indicate the proportion of each genotype.

**Table 6. Correlations Between Six Genetic Indicators and the Knockdown Resistance Mutations in Different Clustered *Ae.albopictus* Populations.**

| Population | NA | NE | I | Ho | He | F(Calculate the power of the natural logarithm base) | N |
|---|---|---|---|---|---|---|---|
| SM | 9 | 6.52 | 2.01 | 0.93 | 0.85 | 0.90 | 30 |
| FZ | 11 | 7.26 | 2.17 | 1.00 | 0.86 | 0.85 | 36 |
| PT | 10 | 7.26 | 2.12 | 1.00 | 0.86 | 0.85 | 30 |
| QZ | 10 | 6.34 | 2.04 | 1.00 | 0.84 | 0.83 | 26 |
| ZZ | 12 | 8.18 | 2.28 | 1.00 | 0.88 | 0.87 | 32 |
| Correlation coefficient | 0.51 | 0.58 | 0.58 | 0.12 | 0.62 | 0.23 | |
| **p-value** | **0.38** | **0.31** | **0.30** | **0.84** | **0.27** | **0.72** | |

Note: The fixation index is negative and is analyzed after doing the natural base power conversion.

**Table 7. Correlations Between Six Genetic Indicators and Dengue Incidence in Different Clustered *Ae.albopictus* Populations.**

| Group | NA | NE | I | Ho | He | Fis (Calculate the power of the natural logarithm base) | Dengue fever incidence rate |
|---|---|---|---|---|---|---|---|
| 1 | 17 | 8.85 | 2.42 | 0.98 | 0.89 | 0.89 | 4.10 |
| 2 | 14 | 8.38 | 2.35 | 1.00 | 0.88 | 0.87 | 5.33 |
| 3 | 12 | 9.38 | 2.34 | 1.00 | 0.89 | 0.89 | 2.76 |
| 4 | 10 | 7.90 | 2.16 | 1.00 | 0.87 | 0.87 | 1.56 |
| Correlation coefficient | 0.74 | 0.19 | 0.77 | -0.27 | 0.26 | 0.30 | |
| **p-value** | **0.26** | **0.83** | **0.23** | **0.73** | **0.74** | **0.70** | |

Note: Cluster division is based on the results of PCoA; the inbreeding coefficient is negative and is analyzed after doing the natural base power conversion; the incidence rate of dengue fever is queried from China CDC, unit (1/100,000)

observed with minimal population variation, consistent with previous research [5,17,18].The observed heterozygosity exceeded the expected heterozygosity, indicating a high prevalent of heterozygotes within the populations [5,18–20]. Furthermore, PCoA results partitioned the eight populations into four distinct groups, with negative inbreeding coefficients for each group, signifying minimal inbreeding. Genetic divergence among these groups ranged from 0.71% to 1.81%, suggesting moderate to high differentiation. Gene flow, indicated to be greater than 1 for each group, pointed towards significant genetic exchange among populations. AMOVA revealed that the genetic distinctions among *Ae.albopictus* populations were not substantial. The high genetic differentiation observed in our study is likely a result of a combination of geographic isolation, selective pressures, and genetic drift. It might also be caused by factors such as the bottleneck effect in the population's history [21]. While gene flow can counteract genetic divergence, it was not substantial enough in our study to prevent high differentiation. Understanding these processes is crucial for designing effective vector control strategies, especially in regions where genetic divergence might influence the effectiveness of insecticides.

### The relationship between resistance of *Ae.albopictus* to deltamethrin to knockdown resistance mutations

This study found that the PT and FZ regions had the highest levels of deltamethrin resistance, potentially related to local insecticide use, dengue outbreaks, or he minimal genetic difference between *Ae.albopictus* populations in these regions. The two pyrethroid resistance-related alleles I1532I and F1534F, identified in the *VGSC* gene, and several other non-synonymous mutations may reduce the insecticidal effect of deltamethrin by affecting its binding ability to sodium channels. There was no significant difference in the correlation between the mutation rates of I1532 and F1534 and the genetic indicators, indicating that the kdr mutations have not yet had an impact on the genetic level of the population at the mic10, mic12, and mic16 loci. Further in-depth research is needed to determine whether these mutations have an impact on the genetic structure of the population. However, we observed that populations with higher mutation rates were more resistant to pesticides, aligning with the findings of Shan et al [3]. We also observed that populations with greater genetic differences (F) tended to have lower frequencies of F1534 and higher frequencies of I1532I. Yijie Li et al. evaluated the multiple insecticide resistances (IR) in urban *Ae.albopictus* populations. In the *Ae.albopictus* populations in Guangzhou, they found that the F1534S and F1534L mutations were significantly associated with deltamethrin resistance [22]. Besides, long - term insecticide selection and genetic isolation of *Ae.albopictus* populations seem to be the reasons for the continuous, patchy distribution of kdr mutant alleles. The small - scale spatial heterogeneity in the

distribution and frequency of kdr mutations may have important implications for vector control operations and insecticide resistance management strategies [23].

## Correlation between genetic indicators and dengue incidence

Pearson correlation analysis showed no significant link between genetic markers and dengue fever incidence, with *p*-values exceeding the threshold for significance ($p > 0.05$). This lack of correlation could be attributed to the geographical proximity of the sampled populations [5], implying that the spread of dengue fever is likely influenced by a multitude of factors. These factors encompass not only the genetic attributes of the mosquito vector but also encompass environmental conditions, climatic patterns, and human behaviors. The *Ae. albopictus* populations in the southeastern coastal areas of China exhibit a moderate to high level of differentiation. Frequent gene flow is an important factor in preventing differentiation among populations. Meanwhile, gene flow also endows *Ae.albopictus* populations in different geographical regions with similar virus - carrying capabilities [24]. Additionally, as areas with intensive human exchanges, the coastal regions face an increasing pressure from imported cases of mosquito - borne infectious diseases such as dengue fever. This insight underscores the complexity of dengue transmission, highlighting that it cannot be solely attributed to the genetic traits of *Ae.albopictus*.

## Dengue fever prevention and control strategy

Recognizing dengue fever's rapid global spread, the WHO highlighted it as one of the "ten global health threats" in 2019, emphasizing the urgency of controlling infection sources and transmission pathways in the absence of a universally effective vaccine or treatment [25]. The findings of this study emphasize the importance of consistent efforts to manage dengue outbreaks, highlighting the necessity for routine mosquito vector control, tailored to local and temporal conditions, to regulate *Aedes* populations. The *Ae.albopictus* populations in the southeastern coastal areas of China exhibit a medium to high level of differentiation. Frequent gene flow is an important factor in preventing the occurrence of differentiation among populations. At the same time, gene flow also enables the *Ae.albopictus* populations in different geographical regions to have similar abilities to carry viruses. Our study confirmed that *these populations exhibit similar virus-carrying capacities, indicating a potential for widespread transmission of dengue fever*, aligning with the findings of Wei Yong and Guo Song et al [5,21,24]. In other words, we observed that the genetic diversity and population structure of Aedes albopictus in the sampled locations suggest that these populations have comparable capabilities to carry and transmit the dengue virus. This observation is based on the genetic similarity and the absence of significant barriers to gene flow among these populations. Additionally, by selecting numerous mosquito sampling sites and *Ae.albopictus* populations in Fujian Province, this study has enriched the genetic research landscape and addressed a gap in the genetic studies of *Ae.albopictus* populations within the province. The study also suggests that if the environments among these cities are similar and there is frequent human mobility, the occurrence and epidemic of dengue fever in one area may potentially trigger the same in other regions simultaneously [25–27]. The results also imply that the presence of other similar factors could pose a risk for extensive disease spread, offering new insights for epidemic prevention strategies. In addition, our study suggests that the type and appropriate dosage of insecticide should be used rationally in controlling *Ae.albopictus*. Based on the analysis of the resistance of *Ae.albopictus* to deltamethrin and *kdr* mutations of 5 strains in Fujian province, we emphasize the need for targeted and strategic use of insecticides to effectively control the *Ae.albopictus* population while minimizing the development of resistance.

### Directions for future research

The study's findings set a foundation for future research, warranting a deeper exploration of non-genetic factors in dengue transmission and the development of innovative mosquito control strategies. Moreover, there is a pressing need for more comprehensive studies that delve into the intricate dynamics between *Ae.albopictus* and dengue viruses across various geographic locales. Such research is imperative for enhancing the early warning, prevention, and control mechanisms against dengue fever.

## Conclusion

This study systematically evaluated the genetic diversity, population structure, and inter-population relationships of *Ae.albopictus* populations in primary dengue fever surveillance provinces along the southeast coast of China. It also analyzed the correlation between various genetic indicators and the incidence of dengue fever. The populations of *Ae.albopictus* were divided into four groups, characterized by rich individual genetic diversity and low inter-species genetic differences, potentially influenced by human activities and transportation. No significant correlation was found between the genetic diversity of *Ae.albopictus* populations and the incidence of dengue fever, suggesting that the sampled populations have similar virus-carrying capacities from a vector perspective. These findings highlight the importance of vector control and the need for controlling the density of *Aedes* mosquitoes according to local conditions and timing, providing new perspective for epidemic prevention.

## Supporting information

**S1 Fig. Capillary Electrophoresis Peaks at Each Microsatellite Locus of the Fuzhou Strain.**
(TIF)

**S2 Fig. Graph of AMOVA results for the three microsatellite loci.**
(TIF)

**S1 Table. Primer sequences for PCR amplification of microsatellite loci (upstream primer with M13 sequence tag (5' – CCCTCATAGTTAGCGTAACG – 3') [5].**
(DOCX)

**S1 Appendix. Excel. S2 Bioassay Experiments.**
(XLSX)

## Acknowledgments

The authors gratefully acknowledge the generous donation of the sensitive *Ae. albopictus* FS population from Professor Xiaoguang Chen's group at Southern Medical University. We are also deeply indebted to Jing-Wu (Southern Medical University), Dr. Wen-Qiang Yang (Zhuhai Center for Disease Control and Prevention), and Dr. Bin-bin Jin (Hangzhou Center for Disease Control and Prevention) for their invaluable assistance in sample collection.

## Author contributions

**Conceptualization:** Shi-Yuan Lin, Pei-Ling Ye, Nan Zheng, Cheng Wu, Li-hua Xie.

**Data curation:** Shi-Yuan Lin, Pei-Ling Ye, Ya-Hui Chen, Nan Zheng.

**Formal analysis:** Shi-Yuan Lin, Ya-Hui Chen.

**Funding acquisition:** Cheng Wu, Li-hua Xie.

**Investigation:** Shi-Yuan Lin, Pei-Ling Ye, Nan Zheng.

**Methodology:** Shi-Yuan Lin, Pei-Ling Ye, Nan Zheng, Cheng Wu, Li-hua Xie.

**Writing – original draft:** Shi-Yuan Lin.

**Writing – review & editing:** Shi-Yuan Lin, Li-hua Xie.

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
