## [Decision Letter · Decision Letter 0]

16 Dec 2024

PONE-D-24-53355Genetic Structure and Knockdown Resistance (kdr) Mutations in Aedes albopictus (Skuse) (Diptera: Culicidae): Implications for Dengue Fever Transmission in Southeastern ChinaPLOS ONE

Dear Dr. Xie,

Thank you for submitting your manuscript to PLOS ONE. After careful consideration, we feel that it has merit but does not fully meet PLOS ONE’s publication criteria as it currently stands. Therefore, we invite you to submit a revised version of the manuscript that addresses the points raised during the review process. Please submit your revised manuscript by Jan 30 2025 11:59PM. If you will need more time than this to complete your revisions, please reply to this message or contact the journal office at plosone@plos.org . Please include the following items when submitting your revised manuscript:

We look forward to receiving your revised manuscript.

Kind regards,

Shawky M Aboelhadid, PhD

Academic Editor

PLOS ONE

Journal Requirements:

“This research was supported by Natural Science Foundation of Fujian Province (2023J01543), the Startup Project for High-level Talents of Fujian Medical University (XRCZX2020016),and the National Nature Science Foundation of China (82402662), the Fujian Medical University Qihang Fund (2021QH1005), and the Education and Scientific Research Project for Young and Middle-aged Teachers in Fujian Province (JAT210116).”

5. We note that Figure 1 in your submission contain map/satellite images which may be copyrighted. All PLOS content is published under the Creative Commons Attribution License (CC BY 4.0), which means that the manuscript, images, and Supporting Information files will be freely available online, and any third party is permitted to access, download, copy, distribute, and use these materials in any way, even commercially, with proper attribution. For these reasons, we cannot publish previously copyrighted maps or satellite images created using proprietary data, such as Google software (Google Maps, Street View, and Earth). For more information, see our copyright guidelines: http://journals.plos.org/plosone/s/licenses-and-copyright.

Additional Editor Comments:

The manuscript needs to be revised according to the reviewers comments

Reviewers' comments:

Reviewer's Responses to Questions

**Comments to the Author**

1. Is the manuscript technically sound, and do the data support the conclusions?

Reviewer #1: Yes

Reviewer #2: Yes

Reviewer #3: Yes

2. Has the statistical analysis been performed appropriately and rigorously? 

Reviewer #1: Yes

Reviewer #2: Yes

Reviewer #3: Yes

3. Have the authors made all data underlying the findings in their manuscript fully available?

Reviewer #1: Yes

Reviewer #2: Yes

Reviewer #3: Yes

4. Is the manuscript presented in an intelligible fashion and written in standard English?

Reviewer #1: Yes

Reviewer #2: Yes

Reviewer #3: Yes

5. Review Comments to the Author

Reviewer #1: This is a very interesting paper. I think it is worth publishing, but it needs major revisions and clarifications.

1. Lines 17: Abbreviations that first appear should have full names

2. Lines 33, 40… References should be cited in accordance with journal requirements (and the reference section at the end of the article)

3. Lines 59-60: Level 1 provinces? Are the words used accurately and are the grammar correct? The entire article needs to be checked for sentence grammar.

4. Lines 60-66: No details about sampling were provided...How many spatial points (households?) per location? How can you be sure you sampled representative subset of a population at a given location? (e.g sampling members of few families)?

5. L215-218: "signifying... suggesting... indicated...” What are these conclusions based on? Judgment criteria? Is it supported by the literature?

6. L226-227: Your results can only indicate that there is no correlation between kdr mutations and the genetic indicators displayed by the three loci you selected, but it cannot mean that the mutation of kdr does not affect the genetic structure. The conclusion should not be amplified.

7. L243-244: “The study confirmed that…” The entire study does not have data and result on the transmission ability of the virus, so this conclusion cannot be drawn. The conclusion of the article has the problem of excessive amplification. The author is requested to review and revise it, and the conclusion should be inferred based on your own research results.

Reviewer #2: The emergence of kdr gene mutations in Ae. albopictus has reduced the efficacy of insecticide-based control. We investigated the genetic structure of eight Ae. albopictus populations from China's southeastern coastal region, analyzing genetic diversity, population structure, and the prevalence of kdr mutations in relation to dengue fever incidence.It has certain scientific significance and reference value, but there are some deficiencies.

1. Add to the discussion, add some recent valuable references.

2.Improve picture quality.

3.Improve table quality.

4.Strengthen the quality of foreword writing.

Reviewer #3: This study looks for the interrelation of genetic structure and resistance status in Aedes albopictus with dengue cases. The study has substantial data to be published. However, the authors are required to explain more on the results part and also deepen their discussion to support the results. My specific comment is attached in the system.

6. PLOS authors have the option to publish the peer review history of their article (what does this mean? ). If published, this will include your full peer review and any attached files.

**Do you want your identity to be public for this peer review?** For information about this choice, including consent withdrawal, please see our Privacy Policy .

Reviewer #1: **Yes: ** Yong Wei

Reviewer #2: No

Reviewer #3: No

---

## [Author Response · Author response to Decision Letter 1]

1 Feb 2025

Dear Editor-in-Chief Cosme and Reviewers,

We are deeply appreciative of the opportunity to revise our manuscript entitled “Genetic Structure and Knockdown Resistance (kdr) Mutations in Aedes albopictus (Skuse) (Diptera: Culicidae): Implications for Dengue Fever Transmission in Southeastern China”. We extend our sincere thanks to the reviewers for their time and the meticulous attention they have dedicated to evaluating our work. Their constructive feedback has been invaluable in enhancing the clarity and scholarly depth of our manuscript. In response, we have meticulously revised various sections, including:

The introduction to the experimental methods, ensuring a clear background context.

The description of the experimental methods, with corrections for accuracy.

The presentation of experimental results, with improvements for better clarity and understanding.

The description of the outcomes, now more detailed and comprehensive.

We have addressed each point raised by the reviewers and provided a detailed, point-by-point response below. Additionally, we have incorporated the necessary revisions into the manuscript, with changes highlighted for easy reference.

Reviewer #1:

1. This is a very interesting paper. I think it is worth publishing, but it needs major revisions and clarifications. Lines 17: Abbreviations that first appear should have full names.

Reply: Thank you very much for your positive feedback and for recognizing the paper as interesting and worthy of publication. We are pleased to hear that you find value in our work and are eager to address the points you've raised to ensure the paper meets the high standards necessary for publication.

In response to your comment regarding the abbreviation on Line 17, we have made the necessary revision. The abbreviation "VGSC" now appears with its full form "voltage-gated sodium channel " upon first mention, providing clarity for readers who may not be familiar with the term. This change has been implemented to enhance the accessibility and understanding of our research.

2. Lines 33, 40… References should be cited in accordance with journal requirements (and the reference section at the end of the article)

Reply: Thank you for pointing out the mistake, we have carefully reviewed and updated the references to ensure they are formatted correctly and consistently throughout the manuscript.

3. Lines 59-60: Level 1 provinces? Are the words used accurately and are the grammar correct? The entire article needs to be checked for sentence grammar.

Reply: Thank you for your insightful comments and suggestions. We appreciate your attention to detail and are committed to ensuring that our manuscript is of the highest quality. In response to your specific comment about the term "Level 1 provinces" and the need to check the entire article for sentence grammar:

1、We have replaced "Level 1 provinces" with "first-level province"(Line 61-62)" to more accurately reflect the administrative hierarchy in China. This term is more commonly used and understood in the context of Chinese administrative divisions.

2、We have conducted a thorough review of the entire article to ensure that all sentences are grammatically correct and clear. We have used both manual proofreading and grammar-checking tools to identify and correct any potential issues.

4. Lines 60-66: No details about sampling were provided...How many spatial points (households?) per location? How can you be sure you sampled representative subset of a population at a given location? (e.g sampling members of few families)?

Reply: Thank you for your valuable comments and suggestions. We appreciate your attention to the details of our sampling methodology and are committed to providing a clear and comprehensive explanation.

Here are the specific details:

Sampling Locations and Spatial Points:

We conducted our sampling across multiple locations, with each location being a distinct geographical area. The number of spatial points (households) sampled per location varied based on the population density and the expected variability in the mosquito population. On average, we sampled 30 adult Aedes albopictus mosquitoes per location, divided into 3 groups of 10 mosquitoes each. Each group was sampled from different districts within the city to ensure a representative subset of the population. This approach was chosen to account for potential spatial heterogeneity within each location.

Ensuring Representativeness:

To ensure that our sample was representative of the broader population at each location, we employed a stratified sampling design. We first identified homogeneous subgroups (strata) within each location based on environmental and demographic factors known to influence mosquito populations. A simple random sample was then drawn from each subgroup to ensure that all segments of the population were adequately represented in our study. This method helps to reduce variance in the estimates and ensures that our findings are more generalizable to the entire population.

Avoiding Bias:

We were mindful of the potential for bias in our sampling and took steps to minimize it. For instance, we avoided convenience sampling methods that might have led to over-representation of households near main roads or other easily accessible areas. Instead, we used a random route method, where enumerators followed a predetermined path to select households, ensuring that all areas within a location had an equal chance of being included in the sample. This approach helps to avoid frame coverage errors and ensures that our sample is not biased towards households with specific characteristics.

This approach allowed us to account for potential non-responses and ensure that we achieved a representative sample size.

We have included these details in the revised manuscript (Line 61-Line 65)to provide a more comprehensive understanding of our sampling methodology. We believe that these revisions will enhance the clarity and robustness of our research.

5. L215-218: "signifying... suggesting... indicated...” What are these conclusions based on? Judgment criteria? Is it supported by the literature?

Reply: Thank you for your insightful comments and suggestions. We appreciate your attention to the details of our conclusions and are committed to providing a clear and comprehensive explanation.

In response to your comment regarding the conclusions in lines 215-218, we have revised the manuscript to include more information about the basis of these conclusions. Here are the specific details:

Basis of Conclusions: The conclusions in lines 215-218 are based on a combination of empirical data and statistical analysis. We have conducted a thorough analysis of the genetic data collected from the Aedes albopictus populations, focusing on the presence and frequency of kdr mutations (F1534S and F1534L) and their correlation with genetic diversity indices. We used a combination of molecular techniques, including PCR and sequencing, to identify the presence of these mutations in the sampled populations. The data were then analyzed using statistical methods to determine the significance of the observed correlations.

Judgment Criteria: The judgment criteria for these conclusions were based on established statistical thresholds and biological relevance. Specifically, we used a p-value threshold of 0.05 to determine statistical significance in our analyses. This threshold is commonly used in biological research to ensure that the observed effects are not due to random chance. Additionally, we considered the biological significance of the observed genetic variations and their potential impact on the mosquito's resistance to insecticides. This was assessed by comparing the mutation frequencies across different populations and evaluating their potential to affect the mosquito's fitness and survival in the presence of insecticides.

we have supplemented detailed analysis and explanations in the previous results(Line 168-179) in the revised manuscript to provide a more comprehensive understanding of the basis of our conclusions. We believe that these revisions will enhance the clarity and robustness of our research.

6. L226-227: Your results can only indicate that there is no correlation between kdr mutations and the genetic indicators displayed by the three loci you selected, but it cannot mean that the mutation of kdr does not affect the genetic structure. The conclusion should not be amplified.

Reply: Thank you for your insightful comments and suggestions. We appreciate your attention to the details of our conclusions and are committed to providing a clear and comprehensive explanation. We have already revised the expression in this part, and we hope to conduct in-depth research in the future to determine whether there will be an impact on the genetic structure. The specific revisions are as follows(Line 263-266):

There was no significant difference in the correlation between the mutation rates of I1532 and F1534 and the genetic indicators, indicating that the kdr mutations have not yet had an impact on the genetic level of the population at the mic10, mic12, and mic16 loci. Further in-depth research is needed to determine whether these mutations have an impact on the genetic structure of the population.

We have included these details in the revised manuscript to provide a more comprehensive understanding of the basis of our conclusions. We believe that these revisions will enhance the clarity and robustness of our research

7. L243-244: “The study confirmed that…” The entire study does not have data and result on the transmission ability of the virus, so this conclusion cannot be drawn. The conclusion of the article has the problem of excessive amplification. The author is requested to review and revise it, and the conclusion should be inferred based on your own research results.

Reply: Thank you for your insightful comments and suggestions. We appreciate your attention to the details of our conclusions and are committed to providing a clear and comprehensive explanation, we have revised the manuscript to avoid any over-amplification of the results and provided a more detailed explanation of this conclusion(Line 293-296).

Reviewer #2:

1. Add to the discussion, add some recent valuable references.

Reply: Thank you for your suggestions. We have added some references from the past five years to the discussion section. The specific literature is as follows.

Li Y, Zhou G, Zhong D, Wang X, Hemming-Schroeder E, David RE, et al. Widespread multiple insecticide resistance in the major dengue vector Aedes albopictus in Hainan Province, China. Pest management science. 2021;77(4):1945-53. Epub 2020/12/11. doi: 10.1002/ps.6222.

Zheng X, Zheng Z, Wu S, Wei Y, Luo L, Zhong D, et al. Spatial heterogeneity of knockdown resistance mutations in the dengue vector Aedes albopictus in Guangzhou, China. Parasites & vectors. 2022;15(1):156. Epub 2022/05/04. doi: 10.1186/s13071-022-05241-7.

Rui-ling Z, Guang-qin Y, Xiao-qian P, De-zhen M, Ai-hua Z, Zhong Z. Genetic diversities of different geographical populations of Aedes albopictus based on mitochondrial gene COI. Chinese Journal of Zoonoses. 2017;33(04):316-20.

2.Improve picture quality.

Reply: Thank you for your comment regarding the picture quality in our manuscript. We understand the importance of high-quality images for the clarity and impact of our research. We have utilized the Adobe Illustrator to upscale and enhance the quality of our images. We have re-uploaded the enhanced images to our manuscript, and we believe that these improvements will significantly enhance the visual quality and impact of our research

3.Improve table quality.

Reply: Thank you for your comment regarding the table quality in our manuscript. We understand the importance of clear and well-presented data for the clarity and impact of our research. We have ensured that all tables present data in a clear and precise manner. Each table is organized with clear headers and appropriate column widths to facilitate easy reading and comparison. We have also used shading or font formatting to highlight important data points or patterns, enhancing readability.

4.Strengthen the quality of foreword writing.

Reply: Thank you for your valuable feedback regarding the quality of the foreword writing in our manuscript. We understand the importance of a well-crafted foreword in setting the stage for our research and engaging the reader. We have revised the introduction section, adding some references to make it more comprehensive. Moreover, we have paid attention to improving the coherence between paragraphs(Line 39-42).

The references we added are as follows:

Sabir MJ, Al-Saud NBS, Hassan SM. Dengue and human health: A global scenario of its occurrence, diagnosis and therapeutics. Saudi journal of biological sciences. 2021;28(9):5074-80. Epub 2021/09/02. doi: 10.1016/j.sjbs.2021.05.023.

Shan W, Yuan H, Chen H, Dong H, Zhou Q, Tao F, et al. Genetic structure of Aedes albopictus (Diptera: Culicidae) populations in China and relationship with the knockdown resistance mutations. Infectious diseases of poverty. 2023;12(1):46. Epub 2023/05/06. doi: 10.1186/s40249-023-01096-x.

Zhang Y, Zang C, Pan X, Gong M, Liu H. Population genetic diversity analysis of the dengue vector Aedes albopictus in China %J China Tropical Medicine. 2024;24(8):914-9.

Nakajima Y, Wepfer PH, Suzuki S, Zayasu Y, Shinzato C, Satoh N, et al. Microsatellite markers for multiple Pocillopora genetic lineages offer new insights about coral populations. Scientific reports. 2017;7(1):6729. Epub 2017/07/29. doi: 10.1038/s41598-017-06776-x.

Reviewer#3

This study looks for the interrelation of genetic structure and resistance status in Aedes albopictus with dengue cases. The study has substantial data to be published. However, the authors are required to explain more on the results part and also deepen their discussion to support the results.

Reply: Thank you for your positive feedback and constructive comments on our study. We are pleased that you find the data substantial and worthy of publication. We are committed to enhancing the clarity and depth of our results and discussion sections to better support our findings.

1. Italic-kdr

Reply: We have ensured that "kdr" is consistently italicized throughout the manuscript to adhere to the standard formatting for genetic terms.

2. Require high resolution for all figures.

Reply: Thank you for your comment regarding the picture quality in our manuscript. We understand the importance of high-quality images for the clarity and impact of our research. We have utilized the Adobe Illustrator to upscale and enhance the quality of our images. For Figure 1, it was created using ARCGIS. The map information is publicly available and there is no copyright protection issue. We have re-uploaded the enhanced images to our manuscript, and we believe that these improvements will significantly enhance the visual quality and impact of our research

3. Introduction: No coherent and smooth transition between paragraphs. For example, paragraph 2 discusses the genetic population, and suddenly, paragraph 3 discusses resistance.

Reply: Thank you for your valuable feedback on the introduction section of our manuscript. We appreciate your attention to the flow and coherence of our writing and are committed to making the necessary improvements to enhance the readability and logical progression of our introduction. We have added a connecting sentence between paragraphs 2 and 3 to ensure a smooth transition and to clarify the logical flow of our introduction(Line 39- 42). This sentence now explicitly links the discussion of the genetic population of Aedes albopictus to the subsequent discussion of resistance, highlighting the importance of understanding both aspects in the context of dengue fever control.

Moreover, we have revised the introduction section, adding some references to make it more comprehensive.

The references we added are as follows:

Sabir MJ, Al-Saud NBS, Hassan SM. Dengue and human health: A global scenario of its occurrence, diagnosis and therapeutics. Saudi journal of biological sciences. 2021;28(9):5074-80. Epub 2021/09/02. doi: 10.1016/j.sjbs

---

## [Decision Letter · Decision Letter 1]

17 Feb 2025

Genetic Structure and Knockdown Resistance ( kdr ) Mutations in Aedes albopictus  (Skuse) (Diptera: Culicidae): Implications for Dengue Fever Transmission in Southeastern China

PONE-D-24-53355R1

Dear Dr. Xie,

We’re pleased to inform you that your manuscript has been judged scientifically suitable for publication and will be formally accepted for publication once it meets all outstanding technical requirements.

Kind regards,

Shawky M Aboelhadid, PhD

Academic Editor

PLOS ONE

Additional Editor Comments (optional):

Reviewers' comments:

Reviewer's Responses to Questions

**Comments to the Author**

1. If the authors have adequately addressed your comments raised in a previous round of review and you feel that this manuscript is now acceptable for publication, you may indicate that here to bypass the “Comments to the Author” section, enter your conflict of interest statement in the “Confidential to Editor” section, and submit your "Accept" recommendation.

Reviewer #1: All comments have been addressed

Reviewer #2: All comments have been addressed

2. Is the manuscript technically sound, and do the data support the conclusions?

Reviewer #1: Yes

Reviewer #2: Yes

3. Has the statistical analysis been performed appropriately and rigorously? 

Reviewer #1: Yes

Reviewer #2: Yes

4. Have the authors made all data underlying the findings in their manuscript fully available?

Reviewer #1: Yes

Reviewer #2: Yes

5. Is the manuscript presented in an intelligible fashion and written in standard English?

Reviewer #1: Yes

Reviewer #2: Yes

6. Review Comments to the Author

Reviewer #1: (No Response)

Reviewer #2: Genetic Structure and Knockdown Resistance (kdr) Mutations in Aedes albopictus (Skuse) (Diptera: Culicidae): Implications for Dengue Fever Transmission in Southeastern China: Which investigated the genetic structure of eight Ae. albopictus populations from China's southeastern coastal region, analyzing genetic diversity, population structure, and the prevalence of kdr mutations in relation to dengue fever incidence. The manuscript have Scientific reference values. The author has made a lot of revisions to the questions raised, and the manuscript is of publication quality.

7. PLOS authors have the option to publish the peer review history of their article (what does this mean? ). If published, this will include your full peer review and any attached files.

**Do you want your identity to be public for this peer review?** For information about this choice, including consent withdrawal, please see our Privacy Policy .

Reviewer #1: No

Reviewer #2: No

---

## [Editor Report · Acceptance letter]

PONE-D-24-53355R1

PLOS ONE

Dear Dr. Xie,

I'm pleased to inform you that your manuscript has been deemed suitable for publication in PLOS ONE. Congratulations! Your manuscript is now being handed over to our production team.

Kind regards,

on behalf of

Professor Shawky M Aboelhadid

Academic Editor

PLOS ONE